# Wireless Passive LC Temperature and Strain Dual-Parameter Sensor

**DOI:** 10.3390/mi12010034

**Published:** 2020-12-30

**Authors:** Ya Wang, Qiulin Tan, Lei Zhang, Baimao Lin, Meipu Li, Zhihong Fan

**Affiliations:** Science and Technology on Electronic Test and Measurement Laboratory, North University of China,Taiyuan 030051, China; wangya0209@163.com (Y.W.); 18734136023@163.com (L.Z.); linbaimao163@163.com (B.L.); 18434367809@163.com (M.L.); 18435131963@163.com (Z.F.)

**Keywords:** temperature-strain sensor, wireless, rotating system

## Abstract

There is an increasing demand for bearing temperature and strain monitoring in high-speed rotating systems. This study proposes a new multiresonance, multiplexing, wireless, passive inductance capacitance (LC) temperature and strain sensor. The sensor has two capacitors connected at different locations (turns) on the same inductor to achieve simultaneous temperature and strain measurements. The plate capacitor is connected to the inner part of the inductor and the other interdigital capacitor is connected to the outer part of the inductor to form two LC loops. The structure of the sensor is optimized through High Frequency Structure Simulator (HFSS) simulations to realize frequency separation of the two parameters and avoid mutual interference between the two signals. The sensor is fabricated on a polyimide film using electroplating technology. The experimental results show that the temperature–strain sensor can operate stably from 25 °C to 85 °C with an average sensitivity of 27.3 kHz/°C within this temperature range. The sensor can detect strains in the range of 1000–5000 με with a strain sensitivity of 100 Hz/με at 25 °C. Therefore, the proposed wireless passive LC temperature-strain sensor exhibits stable performance. In addition, the use of a single inductor effectively reduces the sensor’s area. The flexible substrate provides advantageous surface conformal attachment characteristics suitable for monitoring high-temperature rotating parts in adverse environments.

## 1. Introduction

There is an increasing demand for monitoring temperature, pressure, strain, and other parameters in high-speed rotating environments [1,2,3,4]. Rolling bearings are extensively used in various mechanical devices and precision instruments. During high-speed rotation of the bearing, the temperature of the bearing can increase abruptly due to friction, affecting the normal operation of the device [5]. In addition, the structure of the bearing will be gradually deformed and damaged depending on the action of cyclic loading, leading to bearing failure or even fracture in severe cases [6]. Therefore, it is of great significance to develop sensors for temperature and strain measurements of bearings to determine the safety and stability of engineering components. In recent years, a variety of temperature [7,8,9,10,11] and strain sensors [12,13,14] have been reported. For instance, the thermometers proposed by Rodriguez and Jia [15] are mainly used for temperature detection of rotating bearings. Yang et al. [16] studied a super-sensitive wearable temperature sensor made of graphene nanowalls (GNWs), which can monitor body temperature in real time. The strain sensors proposed by Jia et al. [17] and Fassler and Majidi [18] detect strength and fracture based on plane strain. Mattmann et al. [19] studied sensors for measuring textile strain. However, these sensors measure only a single physical quantity, which is limiting. In practice, it is usually necessary to measure multiple parameters simultaneously [20,21,22]. For instance, the bearing temperature and strain should be monitored simultaneously to monitor the running condition of rolling bearings and predict accidents in advance. Performing measurements of multiple parameters using one sensor can effectively reduce the volume and weight of the measurement system and minimize cost. Many scholars have conducted studies on integrated sensors [23,24,25,26]. LC resonators can be used to monitor dual parameters by superimposing two separate LC resonators. Dong et al. [27] proposed a wireless, passive, LC capacitive temperature-pressure dual-parameter sensor. Tan et al. [22] developed an integrated temperature-pressure-humidity sensor. However, in the above reports, the method of stacking was mostly used to develop multiparametric sensors, which can significantly reduce the sensitivity of the sensor to strain and increase its area.

In order to meet the testing requirements of temperature, strain and other parameters under the environment of rotating bearings. At the same time, in order to realize the effective acquisition of signals in rotation and other environments, this study proposes a wireless passive flexible temperature and strain sensor with multiresonance multiplexing. We used two capacitors connected at different parts of an inductor (different turns) to ensure simultaneous measurement of temperature and strain. Due to the single inductance adopted in this study, an integrated design of the sensor chip is realized, the area of the sensor is effectively reduced, and interaction of the superimposed inductance is also avoided. The sensor is suitable for measuring parameters in narrow and curved spaces. Therefore, the proposed sensor is of great significance for real-time in situ wireless monitoring of all parameters in rotating environments, such as rotating bearings. In this study, the equivalent circuit of the sensor was analyzed, and High Frequency Structure Simulator (HFSS) electromagnetic simulation software was used for simulation analyses of the multiparametric sensor, resulting in the structural optimization of the sensor’s design. This ensured that an integrated dual-parameter sensor with frequency separation of the two sensitive parameters was achieved. The temperature and strain characteristics of the sensor were investigated, and the test results were analyzed. Finally, a precise test was conducted for the temperature and strain parameters.

## 2. Design and Simulation

The principle of the wireless, passive, LC temperature and strain sensor is shown in Figure 1a. The reading antenna was connected to an Agilent network analyzer (E5061B, Agilent, California, USA). The reading circuit includes an equivalent built-in resistance (*R_a_*), a reading antenna inductance (*L_a_*), and an equivalent capacitance (*C_a_*). The sensor circuit is composed of an equivalent resistance (*R_i_*), an equivalent inductance (*L_i_*), and a variable capacitance (*C_i_*) (the index *i* is the abbreviation of the temperature and strain of the sensor). The resonant circuit can be considered to be equivalent to two LC loops, as shown in Figure 1a. The resonant frequency is calculated using Equation (1) [28,29]:(1)$$f_{i}=\frac{1}{2\pi \sqrt{L_{i}C_{i}}},$$
where *f_i_* is the resonant frequency of the sensor’s circuit. It can be observed from the equation that *f_i_* is mainly determined by the sensor’s interdigital capacitance *C_i_*. Interdigital capacitance is a combination of two sets of electrodes intersecting in a comb tooth shape, and the electric field between the interdigital electrodes can be approximated as a uniform electric field [30]. When the number of interdigital electrode pairs is n, the plane interdigital geometric ratio *ŋ* can be defined as,
(2)$$\eta =\frac{g_{c}}{w_{c}+g_{c}},$$
where *g_c_* is the interdigital capacitance spacing and *w_c_* is the interdigital width. When n > 3, the capacitance value can be expressed as:(3)$$C\approx l_{c}(n-1)\varepsilon_{0}\frac{1+\varepsilon_{r}K[{\left(1-\eta^{2}\right)}^{\frac{1}{2}}]}{2K[\eta ]},$$

The *K(x)* function is the complete elliptic integral of the first type [31], n is the interdigital logarithm, ε_0_ is the vacuum permittivity, ε_r_ is the relative permittivity, and *l_c_* is the length of the interdigital electrode. It can be observed from the equation that changes in the permittivity of the substrate, the finger length, finger width, and interfinger spacing of the interdigital capacitor will result in changes in the interdigital capacitance value, thereby changing the resonant frequency.

A parallel plate capacitor is composed of two parallel metal conductor plates, separated by a dielectric material. The capacitance value of the plate capacitor is related to the dielectric, the distance and the facing area between the two plates. Its calculation equation is as follows:(4)$$C=\frac{\varepsilon_{0}\varepsilon_{r}S}{d},$$ where, *ε*_0_ is the vacuum permittivity, and *ε_r_* is the relative permittivity of dielectric. *S* is the area of the capacitor plates, and d is the distance between the upper and lower capacitor plates. The capacitance value changes when *ε_r_* changes.

The sensing mechanism of the sensor is as follows: The antenna is used as the excitation source, the sensor is used as the load of the antenna, the antenna sends out a frequency sweep signal, and the sensor loop is inductively coupled with an interrogation antenna. When the frequency of the sweep signal is close to the natural frequency of the sensor, the sensor’s loop resonates. Sensitive parametric information is fed back to the test antenna [32,33,34]. By extracting the lowest point of the *f–S*_11_ curve, the resonant frequency of the sensor can be obtained. In an LC resonant circuit, any change in the interdigital capacitance or plate capacitor will result in a change in the resonant frequency of the sensor. Changes in the variable capacitance can be determined by observing the changes in the resonant frequency, as given in Equation (1). Accordingly, it has been established that changes in the external parameters are associated with the relationship that determines the physical parameters that lead to changes in the capacitance.

To verify the feasibility of the proposed dual-parameter sensor, we used high-frequency simulation software HFSS to simulate the designed sensor. The antenna was used to feed the sensor by adding an excitation source. The sensor model and simulation results are shown in Figure 1b,c. Figure 1b shows the *f*–S_11_ curve of the sensor. The sensor can detect two resonant frequencies: the temperature signal at approximately 60 MHz and the strain monitoring signal at approximately 120 MHz. Figure 1c shows the electric field distribution diagram of the integrated sensor. As indicated, the electric field distribution of the sensor is mainly concentrated in the electrode part. It can be observed from the simulation that the sensor designed in this study is feasible.

## 3. Results and Discussion

### 3.1. Sensor Preparation

The preparation process of the sensor is shown in Figure 2. First, the polyimide (PI) film was treated with oxygen plasma [32,35] for three minutes. Additionally, a plasma power of 120 W and an O_2_ flow rate of 150 sccm were used to make the film surface hydrophilic. Upper and lower through-holes were drilled on the PI film using laser drilling technology. The outer diameters of the through-holes were 0.4 mm. Silver paste was injected to metalize the through-hole so that the upper and lower surfaces can be connected. We then used electroplating technology to deposit 18 µm of metallic copper on the back of the PI film as a trace that connects the inductor and the capacitor and the lower plate of the plate capacitance. Subsequently, 18 µm of metal copper was deposited on the front of the PI film as the inductor and interdigital electrodes and the upper plate of the plate capacitance. The interdigital capacitor of the strain-sensitive unit was placed on the outer part of the inductor, and the plate capacitor of the temperature-sensitive unit was placed on the inner part of the inductor. The surface of the sensor was covered with the PI film to form an insulating layer, achieving a dual-parameter sensor with multiple resonances. The sensor has the following characteristics: external dimension: 16 mm × 26 mm, size of interdigital electrode: 8 mm × 8 mm, finger length: 6.5 mm, linewidth: 0.1 mm, interdigital distance: 0.2 mm, and number of interdigital pairs: 14, size of plate capacitance: 5mm× 5mm. The outermost length of the inductor is 14.7 mm, with a total of 11 turns. The internal temperature capacitor was connected to a six-turn inductor, and the outer strain capacitor was connected to a five-turn inductor. 

### 3.2. Strain Test

To verify the strain characteristics of the sensor, we built a strain test platform. The cantilever beam device shown in Figure 3a was used to study the strain-sensitive characteristics of the sensor. The cantilever (made of acrylic board) was fixed on a desk, and its other end was hung in air. Pressure was then applied using a pressure gauge to deform the cantilever. The sensor was fixed on the cantilever with glue (number 502, LUSHI, Guangdong Province, China), the temperature-sensitive part was attached to the position where the cantilever did not deform, and the strain-sensitive part was attached to the suspension position of the cantilever so that the sensor can deform together with the cantilever when pressure was applied. A standard strain gauge was fixed at the axisymmetric position of the center of the cantilever beam to detect the deformation of the sensor. Applying pressure at the center axis of the cantilever beam can be equivalent to keeping the strain on the sensor consistent with the strain of the strain gauge [36]. The standard strain gauge was connected to the dynamic strain measuring system (JHDY-0508, Jiangsu, China), and the measured strain was fed back to a computer terminal. Figure 3b shows the change in the strain detected by the strain measuring instrument over time. The result of the one-to-one correspondence between the pressure and strain is shown in Figure 3c. As shown, the strain varied from 0 to 5000με with the applied pressure. First, the strain-sensitive interdigital capacitance was tested using an LCR tester (IM3536, Shanghai, China). The results are shown in Figure 3d. As indicated, the capacitance value decreases as a function of strain from 4.82535 pF to 4.73925 pF. This change is related to the direction of the applied strain [17]. The direction of the applied strain on the capacitor can be mainly divided into two: the direction perpendicular to the interdigital electrode plate and the direction parallel to the length of the interdigital electrode plate, as shown in Figure 4a. When strain is applied in the X direction, the electrode width *w_c_* and electrode spacing *g_c_* are changed, while the electrode length *l_c_* of the interdigital electrode remains unchanged. When strain is applied in the Y direction, the main changes are the electrode length *l_c_* of the interdigital electrode, while the electrode width *w_c_* and the electrode spacing *g_c_* remain unchanged. In this study, the sensor was subjected to strain in the X direction. According to the capacitance calculation formula given in Equation (3), when the distance between the electrode spacing *g_c_* increases, the capacitance value decreases, which is consistent with the detection outcomes.

The sensor was then tested at 25 °C using a vector network analyzer. The number of sampling points of the network analyzer was set to a maximum of 1601 points. A homemade antenna connected to the network analyzer was fixed above the sensor. The test distance between the two was set to 2 mm to ensure that the sensor has a good signal. The sensor strain values were 1000, 2000, 3000, 4000, and 5000 με following the application of forces of 10, 20, 30, 40, and 50 N, respectively, on the central axis of the cantilever beam. Accordingly, the data were observed and recorded. The origin software was used to plot the collected data to generate the *f*–S_11_ response curve of the dual-parameter sensor, as shown in Figure 4. Figure 4a shows the strain mechanism diagram. As shown in Figure 4b, when the strain parameter changes, the temperature-sensitive part of the sensor hardly changes, and the frequency remains essentially unchanged; however, the frequency of the strain-sensitive part changes. The main reason for this result is attributed to the fact that when strain develops (Equation (1)), the value of the strain-sensitive capacitance decreases, leading to an increase in the resonant frequency of the sensor, as shown in Figure 4c. The Figure 4c,d clearly shows the influence of each sensitive unit when the strain changes. The resonant frequency values of each sensitive unit were extracted separately and fitted to obtain the results, as shown in Figure 4e,f. As shown in Figure 4e, the resonant frequency of the strain-sensitive element increases from 121.4 MHz to 121.9 MHz, and the fitted curve is f1 = 1.214 E8 + 100x. Thus, its strain sensitivity is 100 Hz/με.

### 3.3. Temperature Test

The sensor was placed on a heating table to study its temperature-sensitive characteristics. First, an LCR analyzer was used to test the temperature-sensitive capacitor. The temperature plate capacitor was placed on the heating table, and the capacitor was connected to the LCR analyzer using a wire. The heating switch was turned on to heat the plate capacitor; the observations were then recorded. The results are shown in Figure 5a. As indicated, the capacitance increases as a function of temperature. This is attributed to the fact that when the external temperature changes, the plate capacitor medium’s permittivity changes [21,37]. This leads to an increase in the plate capacitance value. When the temperature of the heating platform decreased to 25 °C, the plate capacitor was removed. The sensor was then placed on the heating platform, and the test antenna connected to the network analyzer was used to test the sensor. The sensor was heated from 25 °C to 85 °C, and its generated *f*–S_11_ response curve is shown in Figure 5b. Figure 5b shows that when the temperature parameter changes, the two sensitive parameters of the sensor both change and the frequency shifts to the left. The main reason is that when the temperature increases, the permittivity of the PI film changes; this causes the plate capacitance value to increase from 5.5117 pF to 6.4059 pF. According to Equation (1), the resonant frequency decreases when the variable capacitance increases. As shown in Figure 5c,d, the effects of all the sensitive units during temperature change are clearly observed. The resonant frequency values of each sensitive unit were extracted and fitted to obtain the results, as shown in Figure 5e,f. It can be observed that the resonant frequency varies linearly with temperature. As shown in Figure 5e, the resonant frequency of the temperature-sensitive unit decreases from 63.6 MHz to 61.9 MHz. The fitted curve is f = 6.45E7 – 27,308x, and its temperature sensitivity is 27.30 KHz/°C. Figure 5f shows that the resonant frequency of the strain-sensitive element decreases from 123.97 MHz to 120.65 MHz when subjected to temperature changes, and the fitted curve is f = 1.257 E8 − 56,000x. When the sensor is placed in a temperature-strain compound environment, put the temperature value into the formula to get the effect of temperature on strain, and then get the actual strain value. 

Table 1 presents a comparison of the sensor developed in this study with previously reported sensors [15,36,38,39]. The sensor proposed herein has the following advantages:The strain range is sufficiently large to monitor a relatively larger range of strain changes.Integrated measurement of the sensor is realized, the sensor area is reduced, and the temperature and strain can be monitored simultaneously.The sensor comprises a flexible substrate and has the advantage of conformal attachment on curved surfaces.

## 4. Conclusions

This study proposed a wireless, passive, dual-parameter sensor with multiple resonance multiplexing for the first time. Simultaneous measurement of temperature and strain was achieved by designing two capacitors connected to the same inductor. After HFSS simulation and optimization, two temperature and strain sensors with frequency-separated sensitive parameters were obtained, achieving effective reduction in the sensor area and integration of the sensor chip design. The sensor was prepared on a PI film based on the optimized parameters, and the temperature and strain parameters were tested separately. The effects of different sensitive parameters on each sensitive unit of the sensor were analyzed, and the sensitivity of each parameter was obtained through linear fitting. Experimental test results showed that the multiresonant sensor can operate stably at 25–85 °C and 1000–5000 με conditions, and its temperature and strain sensitivities were 27.3 kHz/°C and 100 Hz/με, respectively. The sensor has broad application prospects for real-time in situ wireless monitoring of parameters on high-speed rotating bearings. To apply the sensor to the rotating bearing and to perform engineering practice, we will focus on improving the strain sensitivity of the sensor in future work, as well as reducing the size of the sensor to make it easier to attach it to the bearing for testing.

## Figures and Tables

**Figure 1 micromachines-12-00034-f001:**
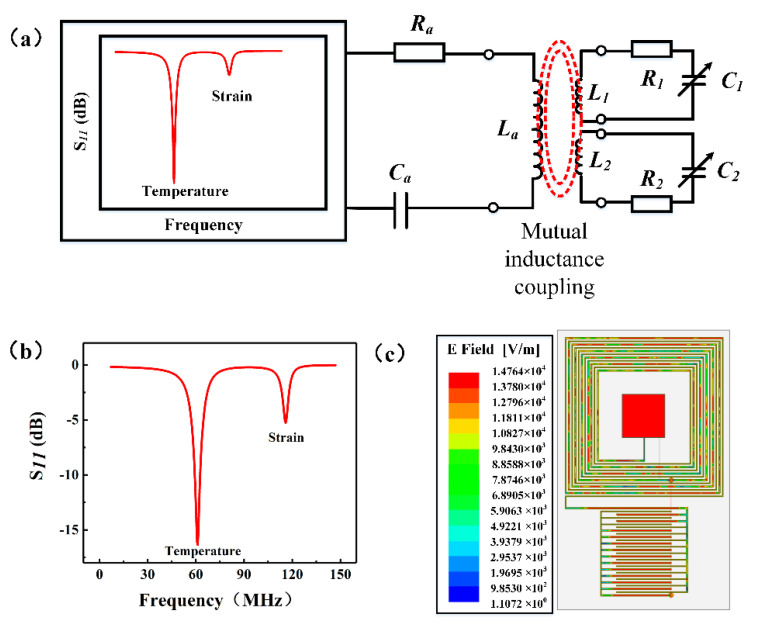
Design and simulation of integrated dual-parameter sensor: (**a**) Sensor’s equivalent circuit diagram; (**b**) Sensor’s *f*–S_11_ simulation curve, and (**c**) Sensor’s simulated electric field distribution.

**Figure 2 micromachines-12-00034-f002:**
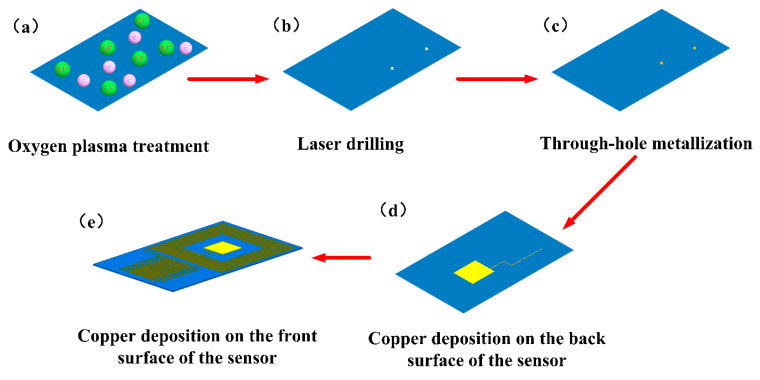
Preparation of integrated dual-parameter sensor: (**a**) Oxygen plasma treatment on the surface of the polyimide (PI) film; (**b**) Laser drilling; (**c**) Through-hole metallization; (**d**) Plating of copper on the posterior part of the PI film; (**e**) Copper deposition on the front part of the PI film.

**Figure 3 micromachines-12-00034-f003:**
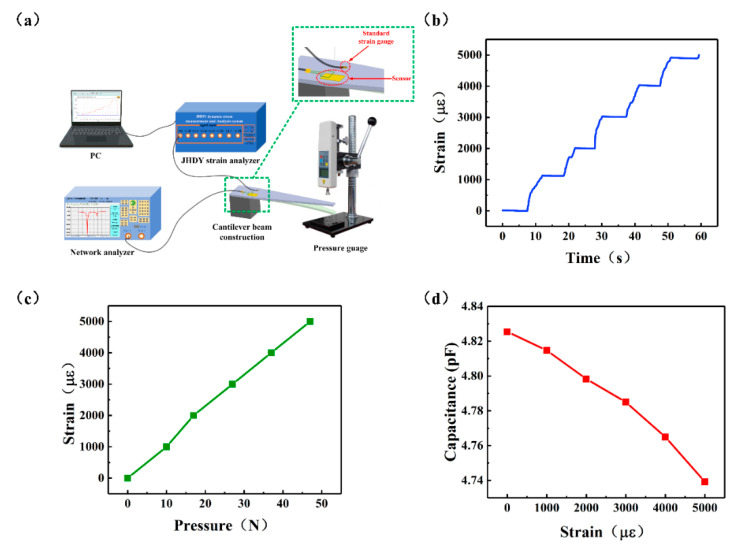
Strain measurement device and results: (**a**) Strain measurement device; (**b**) Experiment data under different pressure; (**c**) Change in strain as a function of pressure, and (**d**) Interdigital capacitance change as a function of strain.

**Figure 4 micromachines-12-00034-f004:**
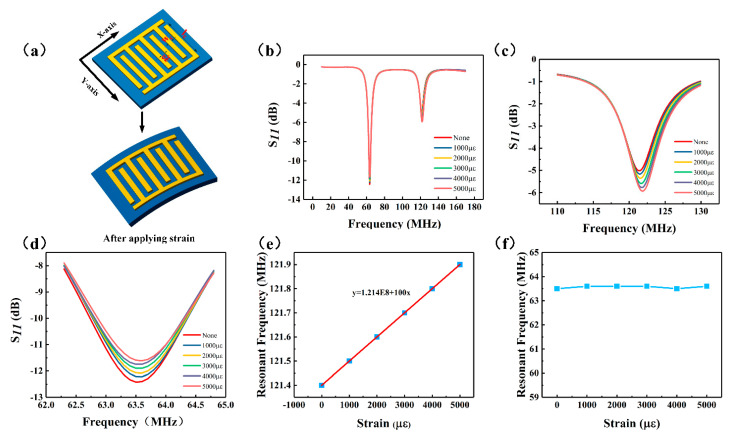
Strain results of the dual-parameter wireless sensor: (**a**) Diagram used to explain the strain mechanism; (**b**) 0–5000 με strain response curve of the dual-parameter sensor; (**c**) Enlarged view of the strain response curve of the strain-sensitive unit; (**d**) Enlarged view of the strain response curve of the temperature-sensitive unit; (**e**) Fitted curve of the strain response of the strain-sensitive unit, and (**f**) Fitted curve of the strain response of the temperature-sensitive unit.

**Figure 5 micromachines-12-00034-f005:**
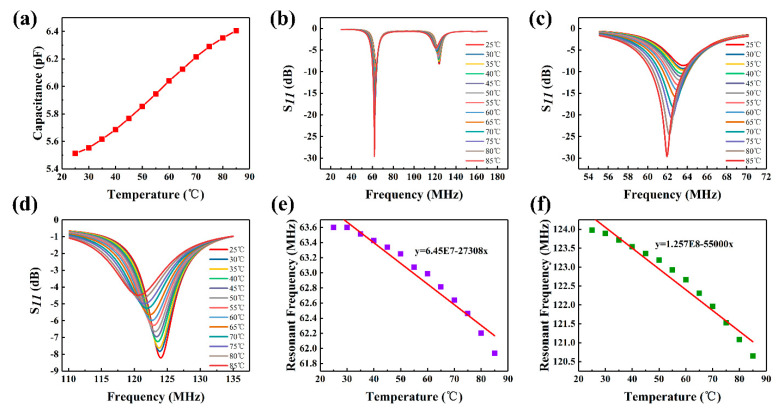
Temperature results of the dual-parameter wireless sensor: (**a**) Plate capacitance change as a function of temperature; (**b**) Temperature response curve (25–85 °C) of the dual-parameter sensor, and an (**c**) enlarged view of the temperature response curve of the temperature-sensitive unit. (**d**) Enlarged view of the temperature response curve of the strain-sensitive unit; (**e**) fitted curve of the temperature response curve of the temperature-sensitive unit, and (**f**) Fitted curve of the temperature response of the strain-sensitive unit.

**Table 1 micromachines-12-00034-t001:** Comparison between the sensors we studied and previously reported sensors.

Sensor Type	Range	Integration	Basal	References
Temperature sensor	0–200 °C	No	rigid	[15]
Temperature sensor	−70–100 °C	No	rigid	[38]
Strain sensor	0–2500 με	No	rigid	[36]
Strain sensor	0–400 με	No	rigid	[39]
Sensor in this study	25–85 °C/0–5000 με	Yes	flexible

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
