# Peer review of "Wireless Passive LC Temperature and Strain Dual-Parameter Sensor"

_micromachines, 2020, doi:10.3390/mi12010034_

Round 1
Reviewer 1 Report
The authors present a new wireless passive LC temperature and strain sensor with multiresonance multiplexing. A detailed overview of related research is given and the importance of developing sensors that simultaneously measure both temperature and strain is emphasized, thus reducing the weight and volume of the sensor and reducing the price.
The authors present the sensor design, work simulation, and results for a new type of sensor that has the purpose of monitoring parameters on high-speed rotating bearings. The content of the paper is clearly presented, the methods and the obtained results are explained in details.
It would be interesting to find out information about sensor characteristics regarding energy consumption. Please explain: Is it a sustainable type of device? In what way do you plan to collect measured data? What is the plan for your future work?
Language of the paper
English needs to be improved.
e.g. there is a language problem in line 33: … environments requires increasing requirements.
Technical quality of the paper
Line 105-107: Figure 1 and the title should be at the bottom of the page
Line 169: Figure 3 title should be at the bottom of the page.
Line 190: Figure 4 should be at the bottom of the page.
Author Response
The manuscript has been rechecked and appropriate changes have been made in accordance with the reviewers’ comments. The revised manuscript was also checked by a native English speaker.
Please see the attachment.

Reviewer 2 Report
The introduction must be improved mentioning at least the aim and scope of the study.
The related work should be presented and compared with the presented work.
The presentation of the results must be detailed.
At the moment, the paper should be improved before acceptance.
Author Response

(The authors gave the same response as above.)

Round 2
Reviewer 2 Report
The authors taken into account my previous comments. I only suggest do detail more about the previous studies in a Related Work section, because it is difficult to understand the functionalities of the previous studies.